# A Skeletal Muscle-Centric View on Time-Restricted Feeding and Obesity under Various Metabolic Challenges in Humans and Animals

**DOI:** 10.3390/ijms24010422

**Published:** 2022-12-27

**Authors:** Christopher Livelo, Yiming Guo, Girish C. Melkani

**Affiliations:** Department of Pathology, Division of Molecular and Cellular Pathology, Heersink School of Medicine, University of Alabama at Birmingham, Birmingham, AL 35294, USA

**Keywords:** obesity, time-restricted feeding/feeding, skeletal muscle disorders/aging, energy metabolism, insulin resistance

## Abstract

Nearly 50% of adults will suffer from obesity in the U.S. by 2030. High obesity rates can lead to high economic and healthcare burdens in addition to elevated mortality rates and reduced health span in patients. Emerging data demonstrate that obesity is a multifactorial complex disease with various etiologies including aging, a lifestyle of chronic high-fat diets (HFD), genetic predispositions, and circadian disruption. Time-restricted feeding/eating (TRF; TRE in humans) is an intervention demonstrated by studies to show promise as an effective alternative therapy for ameliorating the effects of obesity and metabolic disease. New studies have recently suggested that TRF/TRE modulates the skeletal muscle which plays a crucial role in metabolism historically observed to be impaired under obesity. Here we discuss recent findings regarding potential mechanisms underlying TRF’s modulation of skeletal muscle function, metabolism, and structure which may shed light on future research related to TRF as a solution to obesity.

## 1. Introduction

Obesity is a complex multifactorial disease involving environmental factors, genetic predispositions, and human behaviors. Obesity has been commonly associated with muscle dysfunction and given the important metabolic roles and contribution to physical activity, maintaining muscle health is key to attenuation and prevention. A possible therapeutic strategy involves the implementation of a behavior intervention known as time-restricted feeding (TRF) which has garnered attention from the scientific community due to its associated benefits in metabolism and attenuation of obesity among others. To fully address muscle dysfunction and exacerbation of obesity, it may be beneficial to study challenge-specific phenotypes leading up to muscle dysfunction and obesity to identify personalized therapeutic targets for obesity. This review encompasses recent literature regarding the effects of different metabolic challenges on muscle function and metabolism. Further, challenge-specific benefits of TRF or time-restricted eating (TRE in humans) in muscle function and metabolism are discussed which may help elucidate their unique mechanisms.

Specifically, PubMed was utilized for literature search using the keywords “time-restricted feeding muscle obesity”, “time-restricted feeding skeletal muscle metabolism”, “time-restricted feeding muscle aging”, and time-restricted feeding obesity”, Literature containing possible mechanistic insights related to TRF under metabolic challenge were included and additional relevant studies under “Similar articles” were also added. Time-restricted feeding studies began around the year 2013, and all TRF studies from 2013 onwards in both human and animal studies were considered. Studies having relevance to TRF/TRE and muscle function with no specific metabolic challenge were not included.

### 1.1. Current Knowledge Regarding the Benefits of Time-Restricted Feeding/Eating and Obesity

TRF or TRE is an intervention that has gained attention in the scientific community due to its potential as a therapeutic alternative in attenuating obesity and metabolic disease [1]. The general principle of TRF/TRE involves a consolidation of daily caloric intake to the active hours or phases of the day with no alteration to total caloric consumption [2,3]. TRF’s known benefits include the attenuation of obesity-associated phenotypes [4] and improved metabolic function [5]. The beneficial effects have been attributed to a realignment of feeding time with the circadian clock and/or potentially through reduction of insulin resistance [2,3,6,7]. However, attention to the potential mechanisms behind TRF regarding skeletal muscle metabolism, structure, and function and its potential role in the management of obesity has just recently emerged. In this review, we discuss recent findings suggesting how TRF may potentially alleviate the effects of obesity through modulation of skeletal muscle function and metabolism and discuss recently published studies.

Classically, obesity is defined as abnormal or excessive fat accumulation or an energy imbalance that impairs an individual’s health, usually accompanied by the deposition of lipids in non-adipose tissues, such as the liver, heart, and skeletal muscle [8,9]. As skeletal muscle (referred to as muscle) represents approximately 40% of total body mass in a healthy individual [10], it is quantitatively the most important site when considering the detrimental impacts of obesity on physical and metabolic activities. Known causes of obesity include aging, a lifestyle of chronic high-fat or calorie diets, genetic predisposition, and circadian disruption (Figure 1) [11,12,13,14]. The aged population continues to be a growing population with over two billion people in the world projected to be older than 60 by 2050 [15]. The prominence of the Western “American” diet, characterized by low consumption of fruits and vegetables and contrastingly high consumption of fat and sodium, are major contributors to obesity [16]. Crucially, genetic factors also play a role in determining people’s response to the obesogenic environment, as twin and family studies have estimated the BMI heritability to be >40% [17]. Circadian disruption has also emerged as a new concern due to an astounding number (16%) of American workers involved in non-daytime working hours [18] indicated by a recent meta-analysis to be positively associated with overweight risk [19,20,21,22,23].

Obesity is commonly associated with increased adiposity, leading to overall muscle-related dysfunction [24]. Further, obesity has been associated with detrimental impacts on the muscle’s metabolic role in maintaining insulin sensitivity, regulating ectopic lipid deposition, and regulating energy balance [9,24,25,26]. Interestingly, imposing TRE on obese or overweight adults with concurrent exercise has been found to help reduce fat mass and increase lean mass [27]. TRF also showed benefits in muscle performance, structure, and metabolism in *Drosophila melanogaster* under genetic and diet-induced obesity conditions [28]. Interestingly, a recent ongoing study in our lab using *Drosophila* supports the idea that various etiologies of obesity may lead to differences in mechanisms of TRF-induced benefits in muscle [29,30]. In light of this, this review aims to help distinguish between various causes of obesity and assess possible differences of mechanisms in TRF attenuation of muscle function to help address obesity effectively in a population inclusive of various metabolic backgrounds.

The development of obesity is complex and multiple different etiologies have been associated with obesity including poor diet quality, genetic predisposition, aging, and circadian disruption among other things [19,23]. Interestingly, differing mechanisms may underlie the progression of obesity unique to the metabolic challenge (e.g., diet, genetics, aging, or circadian rhythm disruption). For example, chronic lifestyles of high-fat diets can induce overconsumption due to differences in appetite suppression [22], while aging can lead to changes in hormonal production and favor sedentary lifestyles [21]. While differences between metabolic challenges are known at the whole body level, differences in mechanisms are now being explored in skeletal muscle function and metabolism. Another point of interest is whether time-restricted interventions lead to challenge-specific mechanisms in skeletal muscle modulation which are protective against obesity.

Current obesity prevention and treatment strategies involve lifestyle changes including calorie restriction and prescription weight-loss medication [31]. These avenues of treatment, however, may come with limited long-term success or include the potential for harmful side effects [32] although some forms of caloric energy restriction have been met with modest success (5–10%) in short-term weight loss [33]. However, other individuals on caloric energy restriction regimens have shown levels of adherence to typically last 1–4 months [34] generally followed by significant weight gain within 1 year [35]. Additionally, anti-obesity drugs which suppress appetite or inhibit fat absorption come with potentially dangerous side effects such as increased cardiovascular and cancer risks among many others [36]. TRE may provide individuals with another viable form of therapy suitable for those unable to adhere to calorie restriction and concerned with the potential side effects of pharmacological therapy.

### 1.2. Aging Linkage to Obesity through Sarcopenia and Mitochondrial Dysfunction in Muscle

A key feature of aging is the gradual loss of total muscle mass and reduction in physical function, which are known characteristics of sarcopenia [37]. The underlying pathophysiology of sarcopenia has been described to be caused by age-related declines in anabolic hormone concentrations in serum such as testosterone, human growth hormone, and insulin-like growth factor-1 which contribute to muscle development, maintenance, and rejuvenation [38,39,40]. Sarcopenia can lead to lower levels of physical activity and impaired metabolism [41,42]. Further, sarcopenia-associated pathology can impact an individual ability to perform physical activities leading to less energy expenditure [43]. Reductions in energy expenditure lead to an energy imbalance and is a hallmark of obesity [44], suggesting that the pathophysiology of sarcopenia can have important implications related to the development of obesity. Also, with age, an accumulation of damaged cells and proteins can occur due to a loss of function in mechanisms involved in protein quality control including autophagy, proteasomal degradation, and chaperone-mediated folding [45]. This can lead to increased amounts of damaged mitochondria [46] and subsequently, impaired metabolism manifested by a decrease in resting metabolic rate and overall dysfunction in mitochondrial bioenergetics and lower ATP production [47]. In human studies, there is an observable loss of mitochondrial respiratory activity in aging human skeletal muscle in otherwise healthy men and women [48,49]. Protein levels of a mitochondrial biogenesis regulator, peroxisome proliferator-activated receptor gamma co-activator 1α (PGC-1α) were also found to be reduced which correlated with slower walking speed in healthy adults [50]. From our lab, a recent study measuring the effects of aging and other metabolic challenges observed that *Drosophila* muscle performance was drastically reduced supporting the idea that aging leads to a reduction of muscle physical ability [28]. Addressing the topics of proteostasis and mitochondrial dysfunction in skeletal muscle within the aging population may alleviate factors leading to sarcopenia and subsequent obesity.

### 1.3. Time-Restricted Dietary Regimens Attenuation of Aging through Regulation of Muscle Mass and Ectopic Lipid Deposition

In humans, TRE is a noteworthy therapeutic strategy in combatting age-related sarcopenia [51]. Protein intake is a primary anabolic stimulus [52] and anabolic response is relatively diminished in elderly individuals compared to young adults [52]. With the assumption that sufficient protein is included in meals, imposing TRE through consolidation of meals leading to larger protein boluses per meal could aid in meeting levels of optimal muscle stimulation for synthesis. While currently there is a lack of evidence in humans investigating the effects of TRE in muscle under aging, a study reported that TRE and concurrent exercise with a group of participants having a mean age of 44 found success in reducing fat mass and increasing lean mass in overweight and obese adults [27]. Contrastingly, a study measuring healthy younger adults found that pacing the intake of protein over a longer window throughout the day after exercise was observed to lead to greater protein synthesis rates [53]. Whether TRE conclusively increases muscle mass under aging remains to be explored however, it is to note that the latter study only measured outcomes in younger healthy adults. Another human study related to time-restricted eating in overweight sedentary adults over 65 years of age found a modest improvement in walking speed [54]. Also, a human study found that TRE also led to improvement in skeletal muscle insulin sensitivity and anabolic sensitivity in relatively young men [55]. Lastly, another study with overweight participants having a mean age of 38 demonstrated that TRE affected the periodicity of metabolites of amino acids [56]. With newly emerging studies, mechanisms related to time-restricted regimens and their effects on skeletal muscle and sarcopenic obesity in undeniably growing. Our lab has not evaluated changes in muscle mass under TRF in flies, however, in a recent study, our lab observed improved muscle performance in aged *Drosophila* under TRF [28]. Aged wild-type flies demonstrated a positive increase in muscle performance indicated by flight and climbing ability which mimicked the muscle performance of healthy controls [28]. Studies measuring the effects of TRF and TRE in aging are relatively new and further research is needed in ascertaining the effects of time-restricted regimens on muscle mass in elderly obese individuals [57].

As mentioned previously, mitochondrial dysfunction can occur due to a lack of removal of damaged mitochondria commonly observed in aging. This can lead to inefficient metabolism of lipids and glucose resulting in higher ectopic lipid deposition found in the muscle. From our study, we found that ectopic lipid deposition measured by lipid droplet area was observed to be significantly decreased under TRF in 3-week-old flies [28]. Overall reduction of ectopic lipid deposition can attenuate anabolic resistance [58] altering muscle synthesis and may suggest that time-restricted regimens can attenuate loss of muscle mass in aging [58].

### 1.4. Chronic High-Fat Diets and Obesity Are Linked through Metabolic Inflexibility and Insulin Resistance

Lifestyles consisting of chronic high-fat diets have been shown to lead to various negative effects on skeletal muscle mass and performance in addition to associated metabolic functions [12,59]. Under healthy conditions, mitochondria utilize glucose and lipids effectively to generate usable energy and maintain cellular homeostasis. However, under HFD conditions, metabolic inflexibility occurs as a result of nutrient overload leading to the inability to switch between sources of fuel effectively for energy production due to competing high levels of both carbohydrates and fatty acids [60,61,62]. Downstream consequences may lead to impaired energy production and accumulation of glucose and lipids [63]. To support this, a recent study found that HFD fed *Drosophila* after 4 days displayed a marked reduction in ATP levels and mitochondrial respiration [61]. This suggests that under HFD, inefficient energy production in mitochondria may stem from metabolic inflexibility. A study from our lab also corroborates this finding in which ATP levels were also found to be reduced in HFD-fed *Drosophila* [29,30]. Relating to mitochondrial impairment and ATP reduction, a study in humans under an HFD where biopsies of human muscle were collected showed the downregulation of genes involved in oxidative phosphorylation [64]. Oxidative phosphorylation is an essential pathway involving oxido–redox reactions that can lead to the production of ATP. As ATP is a currency needed by cells to undergo processes of metabolism and allow contraction of skeletal muscle in physical activity [65], management of energy metabolism may be essential in mediating HFD pathology in muscle and the development of obesity.

Chronic high-fat diets are also commonly associated with insulin resistance [66]. With increases in circulating fatty acids, higher amounts of diacylglycerol (DAG) can lead to the activation of protein kinase C (PKC) known to disrupt the insulin signaling pathway and result in insulin resistance [67]. Interestingly, our previous study demonstrated that an insulin resistance marker measured by gene expression of *Neural Lazarillo* (*NLaz*) [68] was increased under obesogenic conditions including HFD in *Drosophila* muscle [28]. As we have reported [28], studies have hypothesized that insulin resistance in humans is commonly linked with mitochondrial dysfunction implying the importance of maintaining mitochondria function [69,70]. It is to note, however, that other studies have shown contradictory results as to whether mitochondrial dysfunction is actually linked with insulin resistance and further study may be needed to establish this connection [71]. The increase in muscle fat content has been shown to correlate with loss in strength, mobility, metabolism and lead to insulin resistance [72,73]. A study conducted on mice also showed that mice exposed to long-term HFDs exhibited atrophy in different leg muscles of mice in conjunction with a fiber-type shift [12]. Interestingly, fiber type shifts are commonly observed in skeletal muscle in cases of obesity with the shift being towards fast fiber types; slow fiber types are inversely correlated with body fat levels due to differences in oxidative capacity [74]. Altogether, the mentioned studies indicate that within chronic HFD, there are multiple ways in which insulin sensitivity and metabolism are impaired in addition to muscle composition and mass. Interventions that address mitochondrial inflexibility and insulin resistance are promising for alleviating metabolic and muscle impairments found in HFD-induced obesity. Though a high-fat diet has been linked with metabolic inflexibility and insulin resistance, nutritional components and quality of diets such as excess carbohydrates and lack of protein may also lead to similar phenotypes observed in HFD.

### 1.5. Time-Restricted Dietary Regimens May Attenuate the Effects of Chronic High-Fat Diets through the Activation of Energy Metabolism, Decreased Triglyceride Synthesis, and Increased Protein Synthesis

Time-restricted regimen studies in general regarding an HFD have shown positive benefits using animal models in metabolic function and muscle physiology [28,75,76,77,78] and also in humans [79]. For example, a study evaluating the effectiveness of TRF on various diet types including HFD using 12-week-old male wild-type mice demonstrated improvement in overall metabolic parameters including insulin sensitivity, body fat accumulation, inflammation, and weight gain [4].

Of note, a study in HFD mice demonstrated that induction of fasting cycles was effective in limiting mitochondrial damage induced by HFD [78]. In an ongoing study in our lab we found that in HFD under TRF in muscle, the purine cycle, a key energy metabolic pathway in muscle was upregulated, confirmed with metabolite analyses [29,30]. As the purine cycle/metabolism has been associated with maintaining ATP in muscle [80] activation of this pathway via TRF may provide a solution for meeting ATP requirements under metabolic inflexibility in HFD. The significance of increased ATP levels requires validation in future studies but suggests a potential mechanism in how TRF circumnavigates around metabolic inflexibility to bolster energy production. Additionally, our lab measured ATP levels which were reduced under HFD in ALF conditions and were increased under TRF [29,30]. Future studies will be needed in validating the role of ATP in HFD improvement under time-restricted regimens and potentially incorporating multiple time points to assess if there are changes in the rhythmicity of available ATP. The involvement of the purine cycle/metabolism in HFD has also been observed by a previous finding where mice under HFD displayed dampened circadian rhythmicity mostly in purine catabolism [81]. This study was not muscle-specific but may suggest that purine cycles to an extent are interconnected with HFD and TRF intervention. Another study using mice (C57BL6/J) under a western diet found that TRF benefits included an extension of muscle performance, motor coordination, and glucose regulation [82]. Interestingly, this study also observed increased immune function (to sepsis) which could potentially corroborate the finding of increased activation of the purine cycle which plays a role in immune function.

In our ongoing study, TRF intervention across HFD and genetically induced obesity resulted in the downregulation of *diacylglycerol O-acyltransferase 2* (*Dgat2*) in muscle. DGAT2 is a key enzyme involved in triglyceride synthesis implying that TRF may reduce ectopic lipid deposition through the regulation of DGAT2 levels. Interestingly, *Dgat2* is also implicated in having a direct role in insulin resistance [83]. The downregulation of *Dgat2* under TRF suggests a possible mechanism in which TRF mediates insulin resistance under HFD. Interestingly, flies bearing a knockdown of *Dgat2* displayed improved muscle performance suggesting that *Dgat2* may also play a key role in muscle pathology [29,30].

In a mouse study, the rhythm of protein synthesis through mTORC1 and its associated amino acids was increased under TRF [4]. As HFD has been correlated with muscle atrophy, modulation of protein synthesis under TRF is noteworthy as a potential countermeasure to addressing HFD-related muscle loss [12]. Although direct measurements of muscle mass were not made, a study in our lab observed that TRF is beneficial to preserving muscle performance in HFD as seen in *Drosophila* [28]. HFD flies under ad libitum feeding had a significant reduction in muscle performance compared to WT flies while flies under TRF imposition displayed significantly improved muscle performance [28]. Further, TRF improved metabolic parameters including body weight, ectopic triglyceride levels, and insulin sensitivity, and prevented compromised integrity of muscle and mitochondria [28]. Further tests are needed to conclude whether improvement of muscle integrity and mitochondria leads to the improvement seen in metabolic parameters.

### 1.6. Thrifty Genes, a Potential Etiology of Genetic-Induced Obesity

In the past, humans and their predecessors had limited food security as food preservation techniques were not yet established. In order to survive under periods of food shortage, genetic adaptations protective against times of famine arose. Subsequently, this selection pressure may have led to the overrepresentation of genetic variants that promote rapid eating and excessive energy storage [84] coined as the thrifty gene hypothesis [85]. Thrust into the current era of abundant food availability, once historical evolutionary advantages which allowed humans to thrive under limited food security now led to a metabolic disadvantage in the modern scene.

Genetic factors play a crucial role in contributing to weight gain and obesity. A systematic review of BMI heritability from 140,525 twins and 42,968 family members estimated that genetic contributions to BMI were 0.47–0.90 from twin studies and 0.24–0.81 from family studies [17]. With the advent of candidate gene studies, genome-wide linkage studies (GWLS), and genome-wide association studies (GWAS), the discovery of candidate genes and loci associated with obesity risk and BMI has been significantly accelerated. To date, >250 loci in the human genome have been reported linked with BMI and obesity risk [86]. The most-studied gene mutations contributing to obesity are mainly located in the leptin/melanocortin pathway which controls appetite and metabolism and regulates energy balance and homeostasis. Within the leptin/melanocortin pathway, leptin is secreted by adipocytes, crosses the blood-brain barrier, and binds to leptin receptors in two subsets of neurons (NPY/AgRP and POMC/CART) in the hypothalamus. NPY/AgRP and POMC/CART control appetite and produce neuropeptides that activate the family of MCRs (MC1R to MC5R), which play crucial roles in energy balance and homeostasis. In addition, leptin can bind to its receptors on skeletal muscle and exert direct physiological effects on glucose and lipid metabolism [87].

### 1.7. Linkage of Genetic Obesity in Altering Skeletal Muscle Structure, Function, and Metabolism

Although monogenic forms of human obesity exist, most instances of human obesity are polygenic. The variants of polygenic obesity mostly differ from one individual to another, therefore, it is challenging to conduct a population study with subjects sharing similar genetic causes. While limited studies are available on genetic predisposition effects on skeletal muscle in human subjects, genetically obese animal models have given significant insights into the impact of genetic obesity on skeletal muscle structure, function, and metabolism. A detailed overview of genetically obese animal models can be found in Lutz and Woods (2012) [88]. The leptin-null ob/ob mouse which lacks leptin production is one of the most-used monogenic obese mice in obesity research [89,90]. The skeletal muscle of ob/ob mice typically exhibits lower muscle mass, greater fat content, a fiber shift to slow/oxidative fiber type, glucose tolerance, and insulin resistance [91,92]. Proteomics analysis of skeletal muscle showed a higher abundance of proteins involved in lipid metabolism (β oxidation) and mitochondrial function (TCA cycle and OXPHOS) but not in glucose metabolism [92], suggesting lipids could be the primary energy source in ob/ob mouse skeletal muscle. Although it is seemingly against the view of lipid-induced insulin resistance, the altered proteomic profiles did indicate metabolic disturbance and this observation could be a form of mitochondrial adaptations on metabolic inflexibility. Furthermore, a recent study demonstrated that exosome-like vesicles (ELVs) from ob/ob mice skeletal muscle have altered profiles of lipids, proteins, and miRNAs, moreover, can transmit insulin resistance signals among muscle cells [93]. Note that the same research group previously evaluated ELVs from diet-induced-obese mice skeletal muscle, which did not alter insulin-induced Akt phosphorylation in recipient muscle cells [86], demonstrating that the muscle phenotypes and responses to genetic and diet-induced obesity could be distinct from each other.

Multiple *Drosophila* genetically obese models have been utilized in obesity studies with a focus on skeletal muscle, such as loss-of-function *Sk*2, *Ifc*, and *Bmm* mutants [94,95,96]. Brain-specific knockdown of leptin analog in *Drosophila* produces obesity hallmarks [97], however, no skeletal muscle aspects have been examined. A study from our lab has demonstrated that *Sk*2, *Ifc*, and *Bmm* mutant flies display comprised muscle performance, higher triglyceride levels, and increased insulin resistance markers in muscle [28]. Furthermore, the *Sk*2 mutant fly exhibits aberrant intramuscular lipid infiltration, sarcomere disorganization, and mitochondrial deformation [28]. Although mitochondrial function has not been directly measured, a follow-up study has shown upregulated expression levels of ETC genes, accompanied by reduced ATP levels in *Sk*2 fly muscle [29,30], serving as a support to mitochondrial disturbance.

### 1.8. Time-Restricted Dietary Regimen Benefits in a Genetic-Induced Obesity Model (Sphingosine Kinase 2; Sk2)

TRF studies using different genetically obese mice models have shown improvements in physiological and metabolic parameters, however to our knowledge, muscle-specific TRF benefits have either been only suggested or not evaluated in those studies [98,99]. On the other hand, TRF-mediated changes have been demonstrated in genetically obese *Drosophila Sk*2 mutant flies [28]. An intervention of 12 h active-phase TRF can improve muscle performance, suppress ectopic intramuscular fat deposits, attenuate mitochondrial aberrations, reduce markers of insulin resistance [28], and increase ATP levels in muscle [29,30]. Furthermore, time-series muscle transcriptome data have shown distinct upregulation of genes associated with AMPK-signaling pathways including glycolysis, glycogen metabolism, TCA, ETC in genetically obese *Sk*2 fly but not in the high-fat-diet-induced obese fly [29,30]. Rhythmic genes involved in glycolysis and glycogen metabolism have peaked expression levels at the end of the active phase [29,30]. Corroborating our observations, a study looking at MC4RKO obese mice also found significant improvement in substrate and energy metabolism [99] which may suggest modulations in mitochondrial function due to TRF.

Altogether, these results suggest TRF improves mitochondrial integrity and capacity in skeletal muscle when organisms are under genetic predisposition challenges. It is important to note that findings from genetic obesity studies are typically genotype-dependent, and may not hold true in studies conducted with different genetically obese models. Studies of obese subjects carrying different genetic variants may receive non-significant results simply due to the masking effects of genotype-dependent changes. Therefore, extra caution will need to be taken for studies using obese subjects and genotype-informed analysis will potentially provide the opportunity for personalized treatment and precision medicine.

### 1.9. Circadian Disruption and Obesity Linked through Alterations in Mitochondrial and Muscle Function

Circadian rhythms are near 24-h biological cycles that occur in anticipation of the daily changes in the organism’s environment [100]. Rhythms are driven by the circadian clock which are internal oscillators that control circadian rhythms through cell-autonomous transcription-translation-based feedback loops (TTFL) [101]. The well-characterized central clock is located in the suprachiasmatic nucleus (SCN) however, peripheral clocks including the skeletal muscle have only begun to be investigated in the past decade. Recent studies profiling circadian expressed genes revealed that more than 2300 genes in the skeletal muscle were circadian and mostly involved in transcription, lipid metabolism, and signaling in muscle [102]. Maintenance of rhythmic patterns of circadian genes is believed to be beneficial for the organism in allowing successful anticipation of diurnal changes in the environment.

In circadian disruption or specifically in mutant mice (*Bmal*^−/−^), insulin sensitivity and glucose tolerance were impaired [103]. Consistent with another study, where *Bmal*^−/−^ was made specifically in skeletal muscle, glucose intolerance was also observed [104] in addition to reduced glycolytic flux shown from metabolomics analyses [104]. It was also suggested that the muscle was relying on fat as a source of fuel in conjunction with protein breakdown to support the TCA cycle [104]. In a separate study, *Bmal*^−/−^ and *Clock*^−/−^ mice displayed ~40% decrease in mitochondrial volume in skeletal muscle with aberrant morphology and respiratory uncoupling [105]. Also in a study that utilized feeding restricted only to the inactive phase, the daily rhythm in mitochondrial respiration in rat skeletal muscle was abolished compared to active phase-fed mice [106]. These studies indicate that mitochondrial dysfunction is heavily altered in muscle under circadian disruption potentially leading to dysmetabolism.

Another study using muscle-specific *Bmal1* disruption led to a reduction of force production, skeletal muscle glucose uptake, and glucose oxidation [107]. A study from our lab also investigated the effects of circadian disruption in the muscle through the imposition of a constant light/light 24-h cycle which led to a drastic reduction of muscle performance in flight and climbing ability seen in *Drosophila* [28]. Further, through cytological assays, muscle integrity was compromised and further, ectopic lipid deposition was found to be increased [28].

### 1.10. Time-Restricted Dietary Regimens May Attenuate the Effects of Circadian Disruption through the Improvement of Muscle Performance and Metabolism

A feature of circadian rhythms is their ability to be entrained which occurs when the phase of the clock is modulated to be aligned with the timing of a cue such as light [108]. The skeletal muscle can be entrained by the imposition of feeding/fasting cycles and activity, light entrainment can also entrain muscle rhythms indirectly through the central clock in the SCN. As mentioned previously, a study in our lab demonstrated that *Drosophila* under 24-h light cycle displayed reduced muscle performance, muscle integrity, and increased ectopic lipid deposition [28]. Flies under circadian disruption were also evaluated under TRF and found that muscle performance was significantly improved in addition to amelioration in ectopic lipids and muscle integrity [28]. Though these results indicate that TRF is a potential solution in mediating the improvement of muscle under circadian disruption, there is still relatively insufficient information regarding TRF’s effects on circadian disruption and skeletal muscle in other models. There is, however, a study that has examined glucose tolerance, a trait commonly associated with a muscle parameter, where daytime eating has been shown to prevent circadian misalignment and glucose intolerance under conditions of night work [109]. Additionally, a study although focusing on pancreatic β cells, acknowledged that beneficial effects including insulin-mediated glucose uptake were observed, this may have also been attributed to TRF effects in skeletal muscle [110].

## 2. Conclusions

As previously noted, obesity is a complex multifactorial disease that involves components related to environmental factors, genetic predispositions, and human behaviors. Obesity has been commonly associated with muscle dysfunction and given the important metabolic roles and contribution to physical activity, maintaining muscle health is key to attenuation and prevention. To fully address muscle dysfunction and exacerbation of obesity, common hallmarks within various metabolic challenges leading up to muscle dysfunction and obesity should be investigated to identify therapeutic targets for obesity caused by different challenges.

TRF, a novel intervention has garnered the attention of the scientific community due to its observed benefits in multiple different diseases including obesity. TRF provides another alternative therapeutic strategy in addition to approaches such as calorie restriction and pharmaceutical therapy which may not have limitations which include limited long-term adherence and dangerous side effects. In the context of skeletal muscle, many existing studies, regardless of study settings and obesity type, have demonstrated benefits associated with TRF in modulating muscle structure, function, and metabolism (Figure 2). This study evaluates challenge specific benefits of TRF mainly in skeletal muscle function and metabolism which play important roles in managing obesity. Studies have found that TRF under aging can lead to anabolic sensitivity, insulin sensitivity, and improved uptake of amino acids and that TRF can lead to larger protein intake due to consolidated feeding times allowing improved muscle protein synthesis. Additionally, from our own studies, we found that *Drosophila* displayed improved muscle performance and muscle integrity. TRF in HFD displayed improvements in muscle performance, insulin sensitivity, and reduced inflammation and our study found modulations in gene expression relating to the purine cycle, and levels of ATP in addition to improved muscle performance. Studies regarding genetic-induced obesity demonstrated possible mechanisms related to energy and lipid metabolism in addition to mitochondrial function and mitochondrial integrity. A recent study in our lab found that adenosine monophosphate kinase (AMPK) was upregulated in addition to downstream pathways such as TCA, glycogen metabolism, glycolysis, and ETC which was also supported by metabolite analyses [29,30]. Genetic knockdown of essential genes within these pathways led to severe muscle performance impairment and furthermore displayed a loss of TRF benefits [29,30]. These results suggest that energy metabolism is fundamental to muscle and that muscle-related improvements of TRF are seemingly lost upon the knockdown of these genes. As energy metabolism is a candidate mechanism of TRF improvement, we also measured overall ATP levels in HFD and genetic-induced obesity and found that ATP levels were diminished in ALF conditions and increased under TRF [29,30]. Taken together, energy metabolism seems to be essential in the TRF-mediated attenuation of muscle dysfunction and possibly obesity. Currently, regarding circadian disruption, TRF insight is lacking whereby few parameters have been studied. Additional studies are needed to provide more conclusive insight. However, TRF studies do suggest a difference in benefits relating to muscle-related metabolism and performance. The current findings discussed in this review can be found in Table 1.

A ramification of the obesity epidemic is increased incidences of several disorders including compromising skeletal muscle and cardiovascular health. To combat obesity-induced comorbidities, there is an urgent need to explore the interaction between genes and environments. An understanding of the mechanistic basis of disease etiology and progression often supports the design of therapies to either prevent or treat disease. This review provides current information on whether changing the eating pattern to reinforce the mechanisms of daily rhythms in feeding-fasting will improve skeletal muscle health in an animal model and its translational potential. TRF promotes the integration of nutrient-sensing pathways with the circadian oscillator, thereby optimizing healthy metabolism—likely by maintaining striated muscle lipid and mitochondrial homeostasis, sustaining proteostasis, and enhancing insulin sensitivity. The pleiotropic beneficial effects of TRF in animal models in mitigating multiple metabolic without altering nutrition quality or quantity has opened a potential lifestyle modification strategy to combat cardiometabolic and skeletal muscle disorders.

## 3. Future Directions

### 3.1. Considerations for Future TRE/TRF Studies in Interpreting Metabolic Data

Current studies including those on human patients have provided valuable information regarding the beneficial effects of TRE/TRF in metabolism and attenuation of obesity. Most studies, however, only include a single time point rather than taking measurements throughout the 24-hr day in assays measuring insulin sensitivity and ATP measurements. This may limit the interpretation of TRF as shifts in the timing of lunch and dinner could lead to differences in glucose tolerance depending on the time of measurement [111]. Furthermore, with ATP measurements shown to be strongly clock controlled [112], single time point data collection fails to address the importance of ATP rhythmicity and only measures overall ATP level with the specific time point or ZT commonly omitted.

Matching of fasting duration before testing insulin sensitivity and triglyceride measurement can also be considered. Acute fasting has been shown to induce insulin resistance and may be mediated by triglyceride increase potentially underestimating the beneficial effects of TRF [113]. Future studies involving obesity can also consider the incorporation of various metabolic challenges to evaluate whether the benefits of TRF improvement are specific to certain challenges or generally applies to multiple challenges leading to obesity.

### 3.2. The Study of Microbiota, Potential Interest in Elucidating TRF Mechanisms Underlying Beneficial Effects on Muscle and Attenuation of Obesity

Studies have shown that microbiota are highly rhythmic, play highly metabolic roles and contribute to muscle function, and are highly sensitive to feeding/fasting times [114,115,116]. Dysbiosis or alterations to the microbiota function and population have also been observed under obesogenic conditions [117]. The microbiota helps regulate muscle mass, metabolism of lipids and glucose, and digestion of indigestible macronutrients leading to energy production and regulation of inflammation. Bacterial populations have been observed to prevent atrophy and increase muscular strength mediated by microbiota-derived metabolites [118]. Short-chain fatty acids produced by microbiota such as acetate and butyrate also can promote energy metabolism by activating AMPK leading to the reduction of acetyl-CoA carboxylase and malonyl-CoA, leading to oxidation of fatty acids in mitochondria of muscle and liver [119].

Emerging studies suggest that metabolic challenges lead to dysbiosis underlying a mechanism for the activation of inflammation, impaired metabolism, and nutrient absorption and energy production. A study investigating changes in microbiota under aging conditions displays that when biological age is used with adjustment to chronological age, microbial richness is decreased and bacteria associated with frailty was increased [120]. This offers another basis in which aging leads to sarcopenia in elderly individuals. Further, dysbiosis because of aging in humans and animals can activate an immune response leading to microbial-induced inflammation contributing to “inflammaging” which can subsequently lead to muscle wasting and shifts in tissue metabolism [121]. A study investigating the influences of an HFD on microbiota populations suggests that alterations of bacterial populations also lead to harmful changes in energy harvest, storage, and inflammation caused by increased gut permeability [122]. Regarding circadian disruptions, the microbiota undergoes diurnal variation in composition and function leaving it susceptible to changes in circadian rhythmicity. Changes in rhythmicity in the gastrointestinal tract could affect the timing and function of microbiota due to its involvement in mediating temperature, pH, and nutrient availability leading to an adaptive circadian clock yielding advantages towards some bacteria and their function at various times [123]. The microbiota, given its diurnal rhythmicity in function and abundance and changes due to diet composition and timing, suggest that TRF may also be involved in preserving or modulating microbiota populations [124]. It remains to be explored as to which specific populations are modulated under TRF and which populations specifically play a role in muscle and metabolic homeostasis. Further, the mechanistic basis of how microbiota specifically leads to the attenuation of obesity under different challenges should be explored.

Overall, TRE/TRF is an attractive therapeutic strategy for attenuating obesity. The modulation of microbiota may be linked to feeding/fasting-related mechanisms as various feeding/fasting cycles can change bacterial populations and function. The microbiota regarding diversity and richness is influenced by various metabolic challenges which suggest that under obesity, microbiota-related alterations occur. Future studies regarding the microbiota and its role in the improvement of skeletal muscle physiology and subsequent improvement of obesity may be explored.

## Figures and Tables

**Figure 1 ijms-24-00422-f001:**
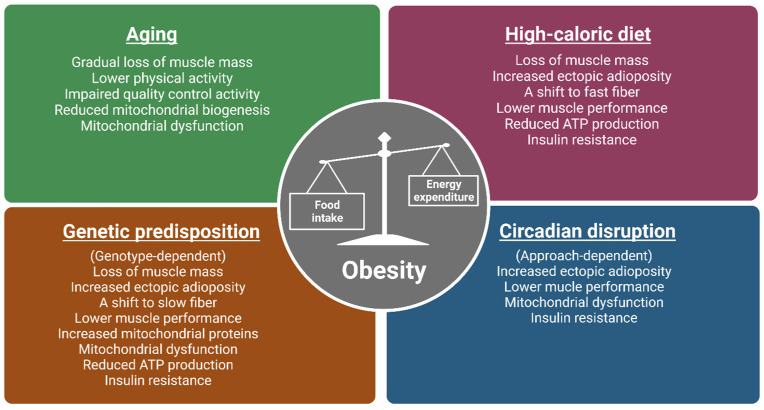
The impacts of various obesogenic challenges on skeletal muscle structure, function, and energy metabolism. Major causes of obesity include aging, a lifestyle of chronic high-calorie diets, genetic predisposition, and circadian disruption. Studies from human and animal models (mice and flies) show that the above-mentioned obesogenic challenges have common and distinct impacts on skeletal muscle structure, function, and metabolism. As muscle phenotypes are genotype-dependent in genetically obese models, only observations from ob/ob, M4RKO mutant mice, and *Sk*2 flies are included. Moreover, as circadian disruption can be induced by light, feeding patterns, and clock gene manipulation, muscle phenotypes can be approach dependent. Images created in Biorender.

**Figure 2 ijms-24-00422-f002:**
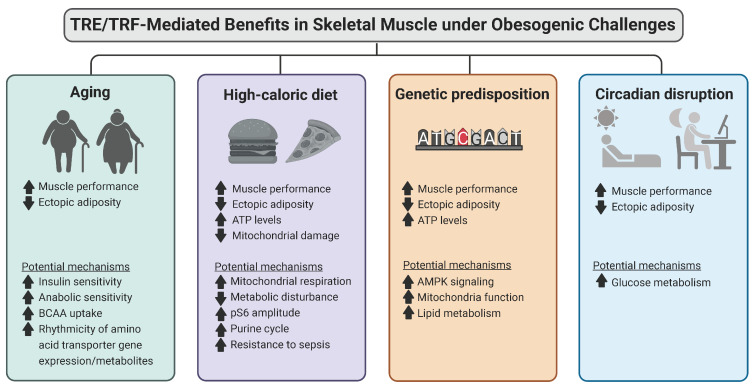
Muscle-specific TRE/TRF-mediated benefits under obesogenic challenges. Studies from human and animal models (mice and flies) show that time-restricted eating/feeding improves muscle structure, function, and metabolism in the context of different obesogenic challenges. Images created in Biorender.

**Table 1 ijms-24-00422-t001:** TRF/TRE benefits found in muscle under different metabolic challenges. Up and down arrows reflect up and down-regulation of indicated parameters respectively under TRF/TRE.

Species	Metabolic Challenge	Observed Phenotypes in Skeletal Muscle	Reference
Human	Aging	↑Insulin sensitivity ↑Anabolic sensitivity ↑BCAA uptake	[55]
Human	Aging/Obesity	↑Rhytmicity of amino acid transporter genes/metabolites	[56]
Human	Aging/Obesity	↑Minor improvement in walking speed	[53]
*Drosophila*	Aging, High-Fat Diet, Circadian Disruption Genetic-obesity	↑Muscle performance ↓Ectopic Lipid Deposition	[28]
Mice, Cells	High-Fat Diet	↓Mitochondrial damage ↓Metabolic disturbance ↑Mitochondrial respiration (metabolic flexibility)	[78]
Mice	High-Fat Diet, Aging	↑Muscle performance, motor coordination ↑Immunity/resistance to sepsis	[82]
Mice	High-Fat Diet	↑Physical endurance ↑Phospho-S6 amplitude	[4]
*Drosophila*	High-Fat Diet	↑Gene expression of purine cycle ↑ATP production ↑Muscle performance	[30]
*Drosophila*	Genetic-obesity	↑Gene expression of AMPK, TCA, ETC, Glycolysis, Glycogen metabolism ↑ATP production ↑Muscle performance ↑p-AMPK	[29]
Mice	Genetic-obesity	Suggestive roles in muscle ↑Muscle mitochondrial fatty acid oxidation ↑Lipid metabolism	[99]
Mice	Circadian disruption	Suggestive roles in muscle ↑Insulin mediated glucose uptake and oxidation	[110]

## Data Availability

Not applicable.

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
