# Peer review of "A Skeletal Muscle-Centric View on Time-Restricted Feeding and Obesity under Various Metabolic Challenges in Humans and Animals"

_ijms, 2022, doi:10.3390/ijms24010422_

Round 1

Reviewer 1 Report

In this paper, “ A skeletal muscle-centric view on time-restricted feeding and obesity under various metabolic challenges in humans and animals” the authors review the current literature on time-restricted feeding and its impact on skeletal muscle function in the context of obesity. The paper is important, and the area covered is niche and warrants a review. Some minor text modifications are required for it to become suitable for publication. I am listing a few examples where the authors can concise their writing and improve the flow of the paper. The authors are advised to do this throughout their manuscript.

Lines 245-248 In order survive 245 under periods of food shortage, genetic adaptations allowing protection against times of 246 famine arose. Please rephrase for clarity

 Line 248 : This 248 was proposed as the thrifty gene hypothesis proposed by James Neel – Please rewrite

Line 418 - It not noted that these TRE/TRF-mediated. Please clarify

Line 423- mechanistic basis?

Lines 436-438. Please rephrase for clarity

Author Response

We appreciate the opportunity to be a part of the IJMS journal. We are grateful for your time and additionally the time of the reviewers in providing their feedback.

With regards to reviewer #1, we are thankful for the kind encouraging words. We have implemented the changes which were advised and are listed below.

Reviewer #1 Comments: “A skeletal muscle-centric view on time-restricted feeding and obesity under various metabolic challenges in humans and animals” the authors review the current literature on time-restricted feeding and its impact on skeletal muscle function in the context of obesity. The paper is important, and the area covered is niche and warrants a review. Some minor text modifications are required for it to become suitable for publication. I am listing a few examples where the authors can concise their writing and improve the flow of the paper. The authors are advised to do this throughout their manuscript.”

Changes have been added throughout the manuscript to aid in conciseness and clarity. Please see the highlighted changes in the revised review.

Lines 245-248 In order survive 245 under periods of food shortage, genetic adaptations allowing protection against times of 246 famine arose. Please rephrase for clarity

Under section 1.6 this sentence has now been replaced with “In order to survive under periods of food shortage, genetic adaptations that were protective against times of famine arose.”

Line 248 : This 248 was proposed as the thrifty gene hypothesis proposed by James Neel – Please rewrite”

Under section 1.6, this sentence was combined with the previous sentence and edited to “Subsequently,

the resulting overrepresentation of genetic variants that promote rapid eating and excessive energy storage

due to selection pressures[84] have been coined as the thrifty gene hypothesis [85].”

Line 418 - It not noted that these TRE/TRF-mediated. Please clarify

This was part of a figure legend from figure 2. The legend now reads “Muscle-specific TRE/TRF-mediated benefits under obesogenic challenges. Studies from human and animal models (mice and flies) show that time-restricted eating/feeding improves muscle structure, function, and metabolism in the context of different obesogenic challenges. Images created in Biorender.

Line 423- mechanistic basis?”

In the conclusion section, this was a typo and has now been corrected to read, “An understanding of the mechanistic basis of disease etiology and progression often supports the design of therapies to either prevent or treat disease”

Lines 436-438. Please rephrase for clarity”

Under the future direction section, this has now been modified to read “Current studies including those on human patients have provided valuable information regarding the beneficial effects of TRE/TRF in metabolism and attenuation of obesity”

Thank you again for your consideration and hope that with these substantial changes, our review can be a part of IJMS.

Reviewer 2 Report

The manuscript presented for review provides an interesting summary of information on time-limited feeding and obesity in various metabolic challenges in humans and animals.

However, the following changes are necessary:

- it is necessary to redraft the text: data on the literature review, what years the publications came from, what databases were searched, what keywords were entered,

- the intention of the authors is not clear to me - the authors reviewed the literature, but in each subsection they refer to their research,

- the introduction itself seems to be insufficient, while the summary is too extensive: it should contain only the most important issues, without citing literature,

- in review publications, summaries in the form of tables, etc. are necessary, which are intended to systematize the information, along with a description of the studied groups, the results obtained - this manuscript definitely lacks table summaries of the individual aspects discussed,

- it is necessary to format the text correctly: title, references.

Author Response

We appreciate the opportunity to be a part of the IJMS journal. We are grateful for your time and additionally the time of the reviewers in providing their feedback.

Regarding reviewer #2, we appreciate the suggestions/comments which were included in the revised review.

Comment #1 “it is necessary to redraft the text: data on the literature review, what years the publications came from, what databases were searched, what keywords were entered

In the second sentence of the introduction, we have now included our inclusion/exclusion criteria of literature used in the review. “Specifically, PubMed was utilized for literature search using the keywords “time-restricted feeding muscle obesity”, “time-restricted feeding skeletal muscle metabolism”, “time-restricted feeding muscle aging”, and time-restricted feeding obesity”, Literature containing possible mechanistic insights related to TRF under metabolic challenge were included and additional relevant studies under “Similar articles” were also added. Time-restricted feeding studies begun around the year 2013, and all TRF studies from 2013 onwards in both human and animal studies were considered. Studies having relevance to TRF/TRE and muscle function with no specific metabolic challenge were not included.”

Comment #2 “the intention of the authors is not clear to me - the authors reviewed the literature, but in each subsection they refer to their research

We appreciate this and agree with this comment. We have reworked the text in all subsections to provide a more holistic presentation of TRF-mediated improvements by adding more discussion about results from other studies. The skeletal muscle benefits associated with TRF is an understudied area therefore, this review aims to discuss recent findings regarding potential mechanisms underlying TRF’s modulation of skeletal muscle function, metabolism, and structure which may shed light on future research related to TRF as a solution to obesity.  We hope this change in text exhibits our intention to discuss the benefits of TRF/TRE in a challenge-specific manner that is inclusive of all recent literature.

Comment #3 “the introduction itself seems to be insufficient, while the summary is too extensive: it should contain only the most important issues, without citing literature”

We have modified the introduction section to include a more in-depth discussion on obesity and its importance in exploring unique metabolic challenges and also the importance of further evaluating TRF/TRE as an intervention that can uniquely benefit the muscle depending on metabolic challenge. Subsections in the introduction have also been modified to include more discussion about other studies regarding TRF-mediated benefits in skeletal muscle under metabolic challenge. Furthermore, we have trimmed the summary to highlight the important issues and removed the majority of literature citations in this section, however, we thought that few of the cited literature were still essential and were retained.

Comment #4 “in review publications, summaries in the form of tables, etc. are necessary, which are intended to systematize the information, along with a description of the studied groups, the results obtained - this manuscript definitely lacks table summaries of the individual aspects discussed”

A table (Table 1) listing the benefits of TRF/TRE has been added along with references to summarize phenotypes observed in skeletal muscle under various challenges.

Comment #5 “it is necessary to format the text correctly: title, references.

We have checked all the references and did not find any specific error with the formatting of text/references. Two of the references (#’s 29 and 30) are included from the conference/symposium and they have a different format than other references.

Thank you again for your consideration and hope that with these substantial changes, our review can be a part of IJMS.